# High Throughput Direct Laser Interference Patterning of Aluminum for Fabrication of Super Hydrophobic Surfaces

**DOI:** 10.3390/ma12091484

**Published:** 2019-05-07

**Authors:** Valentin Lang, Bogdan Voisiat, Andrés Fabián Lasagni

**Affiliations:** 1Institute for Manufacturing Technology, Technische Universität Dresden, George-Bähr-Strasse 3c, 01069 Dresden, Germany; bogdan.voisiat@tu-dresden.de (B.V.); andres_fabian.lasagni@tu-dresden.de (A.F.L.); 2Business Unit Microtechnology, Fraunhofer-Institut für Werkstoff-und Strahltechnik (IWS), Winterbergstraße 28, 01277 Dresden, Germany

**Keywords:** laser interference, surface engineering, manufacturing

## Abstract

This work addresses the fabrication of hydrophobic surface structures by means of direct laser interference patterning using an optical setup optimized for high throughput processing. The developed optical assembly is used to shape the laser beam intensity as well as to obtain the two sub beams required for creating the interference pattern. The resulting beam profile consists of an elongated rectangular laser spot with 5.0 mm × 0.1 mm size, which enables the optimized utilization of the laser fluence available from an ns-pulsed laser with a wavelength of 1064 nm. Depending on the pulse repetition rate applied, heating of the substrate volume generated by heat accumulation encouraged exceptionally high aspect ratios of the trench structures due to melt flow dynamic material deformation. Finally, water contact angle measurements of the produced structures permitted the demonstration of the capability of controlling the wetting angle, in which this effect does not only depend on the height of the generated surface structures but also on their morphology.

## 1. Introduction

When it comes to lightweight construction, aluminum is a commonly used material due to its high strength to weight ratio. Moreover, aluminum offers good corrosion resistance and high thermal conductivity. This material as well as several Al alloys are the most frequently used systems in aircraft construction [1,2,3], and its advantages are further appreciated in automotive [4,5,6], shipbuilding [7,8,9], and apparatus engineering [10,11,12]. The embedding of technical systems in given environmental conditions leads to a functional relation of surfaces to water wettability. The reinforcement of aluminum by intermetallic alloy particles results in a higher sensitivity to local corrosion, making protection against water wetting necessary [13]. Also, water retention on the surface of heat exchangers is problematic as it reduces the air-side heat transfer coefficient and provides a location for biological activity [14]. The aerodynamic disturbance associated with flying in heavy rain is postulated to be one of the causes of storm accidents [15]. In addition, the impact of rain has a direct influence on the fuel efficiency achieved [16]. Moreover, the technical structures of air- and spacecraft are exposed to a risk of ice formation on their outer shell, which can have an effect on the wings’ function [17,18]. Other areas affected by ice formation include power grids, wind turbines, and marine structures in cold climates [19,20,21]. Present scientific approaches show that icephobicity is connected to hydrophobic surface properties [22,23,24]. Previous studies have already demonstrated hydrophobic properties on aluminum by coating an anodized Al_2_O_3_ underlayer with Polytetrafluorethylen (PTFE) [25,26,27]. Further approaches exist in which the implementation of a micro-/nanostructured topography can provide an ice-/hydrophobic character of the surface [28,29,30].

A promising method for producing these micro and nanostructures in a cost-efficient way is direct laser interference patterning (DLIP) [31,32,33]. In DLIP, two or more laser beams are overlapped on the surface of the substrate, creating a periodic intensity distribution. The pattern consists of a high number of high intensity elements (interference maxima) which are produced with a single laser shot. 

In the volume where the coherent laser beams overlap, a periodic modulation of the laser intensity distribution is obtained. Depending on the number of interfering laser beams, the periodic modulation represents linear or dot-like patterns. For two laser beams, a line-like pattern geometry can be obtained and the interference spatial period *Λ* can be calculated as a function of the laser wavelength *λ* and the angle *θ* between the interfering beams using Equation (1):(1)Λ=λ2sin(θ2)

Different optical configurations are available today for obtaining an interference pattern. In all cases, a single intrinsic laser beam is divided into several sub-beams, with different beam splitting techniques available to be used (e.g., beam-splitters [34,35,36], diffractive optical elements [37,38,39], and refractive optics [40]). Then, the irradiated surface is affected locally at the interference maxima positions (e.g., by melting or ablation), allowing the fabrication of the periodic structure. While in laser interference lithography a photoresist material is needed for producing the periodic structure, in direct laser interference patterning the patterns are produced on the substrate surface by selective ablation and/or melting [41]. 

Even though DLIP has already been shown to be a high throughput method compared to other sequential techniques like direct laser writing (DLW), the challenge is not just to be capable of producing these structures with resolution in the micrometer or sub-micrometer range but also with high throughputs.

This work addresses the fabrication of hydrophobic surface structures by means of direct laser interference patterning using an optical setup optimized for high throughput processing. For this aim, a special optical system is used, which realizes both beam shaping and splitting in order to apply the full pulse energy available using an ns laser source. A laser interference setup with elongated elliptical laser spots has been also developed by Molotokaite et al. to produce micro-surface patterns on thin metal films deposited on glass substrates [42]. Moreover, water contact angle measurements of the produced structures are performed to evaluate the possibility of controlling the wettability of these surfaces, taking into consideration morphological changes at the material’s surface. The surface topography of the irradiated metallic substrates is investigated using confocal microscopy (CF) and scanning electron microscopy (SEM).

## 2. Materials and Methods

### 2.1. Materials

The samples used in this work consisted of flat extruded aluminum substrates (EN AW-5754 AlMg3 [43]) with a thickness of 0.9 mm and a surface roughness (Sa) of 270 nm. The substrates were used as received. The properties of the material used are listed in Table 1.

### 2.2. Direct Laser Interference Patterning

Figure 1 shows the modular optical assembly developed for high-throughput DLIP. This device allows the shaping of the form of the laser beam, obtaining rectangular elongated laser spots at the working area on the substrate surface using the combination of a cylindrical telescope and cylindrical lens. In addition, the arrangement of two different bi-prisms permits the splitting of the main laser beam as well as overlapping at the material’s surface. Also, by shifting the cylindrical optical components, the dimensions of the line-shaped laser spot can be tailored to requirements. In our experiments, the main laser spot was shaped into two beams with a dimension of 0.1 × 5.0 mm^2^. The edge angle of the bi-prisms determined the final intersection angle of the sub beams and thus the resulting interference period for a given laser wavelength. Different bi-prisms were used to obtain intersection angles *θ* of 7.6° and 3.1°, which corresponded to interference periods *Λ* of 8.0 µm and 19.0 µm, respectively. The laser source used was a Q-switched diode-pumped slab-type solid-state laser (Nd:YAG) with a fundamental wavelength of 1064 nm and laser pulse duration of 10 ns at repetition rates of up to 10 kHz. The maximum average laser power was 150 W. Since the applied laser energy per pulse was constant (*E_P_* = 16 mJ), the same laser fluence was used in all experiments (*F_P_* = 3.2 J·cm^−2^) where the laser pulse repetition rate and the pulse overlap were varied.

In pulsed laser processing, the ratio between pulse repetition rate *f_rep_* and feed rate *v_f_* has a direct influence on the pulse overlap (PO) of individual pulses and thus has a significant effect on the processing results due to local pulse energy accumulation. According to Equation (2), the pulse overlap is a function of the feed rate *v_f_*, the repetition rate *f_rep_*, and the spot size *l_s_* (in the direction of movement): (2)$$\mathrm{PO}=1-\frac{d_{f}}{l_{s}}=1-\frac{v_{f}}{f_{rep}\cdot l_{s}}$$

### 2.3. Surface Characterization

The characterization of the generated surface topographies was performed using optical microscopy (OM) and SEM. A confocal microscope (SENSOFAR S NEOX, Barcelona, Spain) was used to measure three-dimensional topographies of the produced surfaces. Using a scanning electron microscope (PHILIPS XL30 ESEM, Amsterdam, The Netherlands), high-resolution images of the modified surfaces were acquired. In order to determine the wettability of the processed surfaces, static contact angles with deionized water were performed (KRUESS DSA100S, Hamburg, Germany).

### 2.4. Modelling of Heat Accumulation

Since this work also addresses the accumulation of thermal energy introduced by laser pulses over time, calculations of the temperature evolution at the laser-affected zone were carried out according the model introduced by Weber et al. [44], which can be expressed as
(3)$$T_{Sum,nD}\left(t\right)=\frac{Q_{nD}}{\rho \cdot c_{p}\cdot \sqrt{\left(4\cdot \pi \cdot \kappa \right)}}\cdot \overset{N_{P}}{\underset{N=1}{\sum }}\frac{\Theta \left(t-\frac{N-1}{f_{L}}\right)}{\sqrt{{\left(t-\frac{N-1}{f_{L}}\right)}^{nD}}}\cdot e^{-\frac{1}{\left(t-\frac{N-1}{f_{L}}\right)}\cdot \frac{r_{nD}^{2}}{4\kappa }}$$
where *Q_nD_* is the residual heat introduced into the material, *ρ* is the mass density of the material, *c_p_* is the specific heat capacity, *κ* = *λ_th_*∙(*ρ*∙*c_p_*)^−1^ is the diffusivity, *λ_th_* the heat conductivity, *t* is time, *r* is the coordinate of space, *f_L_* is the laser repetition rate, and *N_P_* is the number of pulses.

Due to the elongated shape of the laser irradiated area, the 1D solution of the heat flow can be assumed (nD = 1). By further setting the volume coordinate *r^2^_nD_* to zero, the following simplified equation is obtained:(4)$$T_{Sum,nD}\left(t\right)=\frac{2\cdot Q_{Heat}/A}{\rho \cdot c_{p}\cdot \sqrt{\left(4\cdot \pi \cdot \kappa \right)}}\cdot \overset{N_{P}}{\underset{N=1}{\sum }}\frac{\Theta \left(t-\frac{N-1}{f_{L}}\right)}{\sqrt{t-\frac{N-1}{f_{L}}}}$$
Here, *Q_Heat_* is the residual heat and *A* is the irradiated area.

In consequence, the temperature at the material surface can be calculated depending on the utilized laser processing parameters.

### 2.5. Modelling of Surface Wettability

The contact angle between a liquid and a solid surface depends on the thermodynamic properties of the system and determines their interaction at the solid-liquid interface. One of the parameters affecting the water contact angle (WCA) is the surfaces roughness. Furthermore, very different WCAs can be measured, depending on the liquid which is in contact with the surface over the entire area and whether cavities filled with air are formed. For the first case, a still widely accepted model was developed by Wenzel in 1936 [45] according to which the WCA can be calculated using [46]
(5)$$\cos\theta_{W}=r\cdot \cos\theta_{Y}$$
where *θ_W_* is the contact angle of the rough surface, *θ_Y_* is the reference contact angle of an imaginary projected flat surface, and *r* is the roughness ratio, which is defined as the ratio of the true rough area of the solid surface to the apparent flat reference area. 

Since *r* is always larger than 1, the apparent contact angle (CA) increases for hydrophobic surfaces and decreases for hydrophilic surfaces. In the case of trench patterns produced with DLIP, the roughness ratio *r* can be described as in Equation (3), where the length of the imaginary cross-sectional longitude *l_y_* corresponds to the interference period *Λ* and the length of the real cross-sectional longitude *l_real_* can be assumed as the sum of the interference period Λ and the structure height *h_st_*:(6)$$\cos\theta_{W,DLIP}=\frac{l_{real}}{l_{Y}}\cdot \cos\theta_{Y}=\frac{\Lambda +2\cdot h_{\mathrm{st}}}{\Lambda }\cdot \cos\theta_{Y}.$$

In the case where the liquid is in contact with both the solid surface and air (including cavities filled with air), the Cassie and Baxter [47] model can be used for calculating the WCA (*θ_CB_*). For the case of liquid covering a fabric made of rounded fibers or sitting on posts that are not flat, a special case of the Cassie-Baxter equation with the addition of a roughness factor (*r_l_*) was established by Marmur [46]:(7)$$\cos\theta_{CB}=r_{l}\cdot f\cdot \cos\theta_{Y}+f-1$$
Here, the coefficient *f* represents the fraction of the solid surface area wet by the liquid.

For the trench (line-like) structures, *f* can be derived from the length *l_co_* of the hills which are in contact with the liquid in relation to the interference period *Λ* (which represents the total apparent area). Hence, the following equation can be utilized:(8)$$\cos\theta_{CB}=r_{l}\cdot \frac{l_{co}}{\Lambda }\cdot \cos\theta_{Y}+\frac{l_{co}}{\Lambda }-1=\frac{\Lambda +2\cdot h_{\mathrm{st}}}{\Lambda }\cdot \frac{l_{co}}{\Lambda }\cdot \cos\theta_{Y}+\frac{l_{co}}{\Lambda }-1$$

## 3. Results and Discussion

The optimized optical design for high-throughput DLIP was used to structure flat aluminum substrates with functional micro trench patterns. Interference periods of 19.0 µm and 8.0 µm were realized by using bi-prisms with different angles as mentioned in the experiential section. Figure 2 shows exemplary scanning electron microscope images of DLIP structured aluminum surfaces with different spatial periods (Figure 2a,b: *Λ* = 19.0 µm; Figure 2c,d: *Λ* = 8.0 µm) which were treated with different pulse overlaps (Figure 2a,c: PO = 80%; Figure 2b,d: PO = 99%). It can be seen that the formed structure has a line-like geometry where the material has been modified at the high intensity peaks of the interference maxima. In particular, the samples treated with high POs (Figure 2b,d) present higher structures compared to those processed at low PO (Figure 2a,c). Also from Figure 2a, it can be seen that when the period is relatively large (*Λ* = 19.0 µm) and the pulse overlap is relatively low (PO = 80%), the material at the interference minima positions remains unaffected. By contrast, for high pulse overlaps (Figure 2b: *Λ* = 19.0 µm, PO = 99%) as well as for the shorter spatial periods (Figure 2c: *Λ* = 8, PO = 80%), the total surface of the material is affected by the laser treatment. This effect can be related to the structure formation mechanism as has already been discussed in [48]. Since the PO is directly correlated with the number of pulses irradiating a certain area (see Equation (9)), high overlaps correspond to a large number of laser pulses (e.g., *N_p_* = 100 for PO = 99%) which can cause increased heating at the material surface:(9)$$N_{p}=\;\frac{1}{1-\mathrm{PO}}.$$

Heating of the material plays a role in the formation of the surface geometry, since the melt flow (Marangoni convection) is a driving principle in structure formation besides recoil and plasma pressure [49,50,51,52,53]. With DLIP, melt flows from the hot interference maxima to the colder interference minima positions. In an initial stage of structure formation, unmelted areas are visible between the melted areas (see Figure 2a). A continuous shift of the melt towards each other leads to a collision of the melt fronts and consequently to a growth in structure height. Since the melt cannot be spread horizontally, the melt volume is pushed upwards (this is visible in Figure 2c). Similar growth in structure height has been previously reported in [54]. In addition, a higher PO introduces an additional roughness in between the trenches produced (sub-micro roughness). One possible explanation involves a random re-deposition of laser ablated material particles. Moreover, an effect of the laser-induced temperature increase on the chemical material composition (e.g., one due to oxidation) is possible.

Figure 3 shows examples of confocal microscope images of the trench structures formed with different interference periods and different POs. With higher POs (e.g., 99%) higher trenches were produced, corresponding to structure heights of 7.6 µm and 32.5 µm for 8.0 µm and 19.0 µm periods, respectively (Figure 3b,d). The height of the periodic structures formed with lower PO (80%) (Figure 3a,c) were 2.2 µm and 2.4 µm for the periods of 8.0 µm and 19.0 µm, respectively. The images also show that the produced structures are homogeneous across the whole examined area. The calculated aspect ratios (AR) (defined as the ratio between the structure height *h_st_* and the spatial period *Λ*) for both periods differ strongly when using the same irradiation conditions. For example, aspect ratios of 0.95 and 1.71 were determined for the treated steel samples with spatial periods of 8.0 µm and 19.0 µm, respectively, at a 99% PO and a laser fluence per pulse of 3.2 J∙cm^−2^ (see Figure 3c,d). This effect can be explained by the melt flow during the structuring process, as recently noted by Lang et al. [48]. In their work it was shown that molten material at the interference maxima positions moves from hotter to colder peripheral areas where it merges at the interference minima locations. The application of subsequent laser pulses, which is in particular more relevant when using high overlaps, results in additional melt leading to patterns with higher aspect ratios. Even though the structure height can be significantly boosted (especially for the larger spatial periods) due to stacking of molten material, a stronger increase of the PO results in a reduction of the structure height for smaller periods due to material overmelting [48].

Since investigations performed for stainless steel have already shown that the resulting structure heights can vary during the continuous DLIP process [40], the structure heights were measured at different positions in the sample, including the starting point (beginning of the laser process or infeed) as well as subsequent positions (up to 50 mm from the starting point).

Figure 4 shows the evolution of structure height over the DLIP process progression for the spatial periods of *Λ* = 19.0 µm (Figure 4a,b) and *Λ* = 8.0 µm (Figure 4c,d). Figure 4a,c show the measured structure height evolutions for different POs between 80% and 99% at constant laser pulse repetition rates of 2 kHz (Figure 4a, *Λ* = 19.0 µm) and 0.1 kHz (Figure 4c, *Λ* = 8.0 µm), respectively. Figure 4b,d display the structure height evolution for pulse repetition rates from 0.1 kHz to 5 kHz for a fixed PO of 98%. Generally, the graphs reveal a logarithmic increase of the structure height during the first 2 mm of the laser process, followed by a steady state condition with a constant value of the structure height. Such a course of the structure height evolution is particularly noticeable with increasing pulse repetition rates. In addition to the PO, the applied pulse repetition rate also affects the resulting structure height. For *Λ*= 19.0 µm, higher pulse repetition rates lead to higher structure heights (Figure 4b). For example, at a repetition rate of 5 kHz an average structure height of 32.5 µm was obtained, while at 0.1 kHz a structure height of only 7.3 µm was reported; in both cases the same laser fluence per pulse (3.2 J∙cm^−2^) and PO (98%, corresponding to Np = 50) were applied.

In the case of the aluminum samples treated with a spatial period *Λ* = 8.0 µm, Figure 4c reveals that the highest structures were not obtained for the highest used PO (99%). For instance, mean structure heights of 7.2 µm and 5.3 µm were observed for a PO of 98% and 98%, respectively. Furthermore, Figure 4d also shows that higher pulse repetition rates result in considerably reduced structure heights for *Λ*= 8.0 µm, compared to the larger spatial periods.

A comparison of the calculated structure height aspect ratios for the trench structures produced with spatial periods of *Λ* = 19.0 µm and *Λ* = 8.0 µm as a function of the PO (and thus also of *N_p_*) as well as the pulse repetition rate, is shown in Figure 5. Significant differences in the AR evolution can be observed. For the larger period *Λ* = 19.0 µm, both an increase in pulse repetition rate and an increase in pulse overlap result in higher aspect ratios. However, as mentioned in the previous section, for the *Λ* = 8.0 µm pattern, the highest AR is obtained for PO = 98% (*N_p_* = 50). A further increase in the PO to 99% results in a reduction of the AR from 0.9 to 0.67. For a higher repetition rate of 5 kHz, the aspect ratio barely varies with change in PO and the maximum value is obtained at PO = 95% (minimum AR = 0.24 at 80% PO; maximum AR = 0.38 at 95% PO). In addition, for the patterns produced with a larger spatial period (*Λ* = 19.0 µm), the maximal calculated AR values are approximately 75% higher compared to structures fabricated with the smaller period (*Λ* = 9.0 µm).

While high pulse repetition rates and high pulse overlap at *Λ* = 19.0 µm led to continuously increasing average structure heights, high pulse repetition rates and high pulse overlap at *Λ* = 8 µm led to decreasing average structure heights. At *Λ* = 19.0 µm, the distances between the intensity maxima of the interference distribution are wide enough for the melt volume to accumulate. By contrast, for *Λ* = 8.0 µm, the distances between the intensity maxima of the interference distribution are smaller, meaning an accumulation of melt volume occurs earlier, followed by an overmelting, which leads to collapse of the structures.

These results indicate that thermal energy provided by the sequence of laser pulses leads to accumulation of energy, resulting in local heating of the substrate volume, which in turn influences the pattern formation.

Figure 6 shows the calculated temperature evolutions at the surface of an aluminum body according to Equation (7). The simulations take into account a laser pulse fluence of 3.2 J∙cm^−2^ with different pulse repetition rates of 0.1, 1, and 5 kHz. The results clearly show the influence of different pulse repetition rates on the local heating of the substrate volume. Despite the relative rapid cooling of the substrate material, a significant increase in the surface temperature of the substrate was calculated even for a pulse repetition rate of only 1 kHz, which can be explained by the accumulation of the thermal energy sequentially introduced by the laser pulses. This increase in the local temperature of the substrate became stronger as higher pulse repetition rates were applied. For example, a temperature increase of 490 K was calculated for a repetition rate of 5 kHz, while increments of 230 K and 50 K were observed for 1 kHz and 0.1 KHz, respectively. The differences between the temperatures at the maxima and minima positions had a significant influence on pattern formation, as has been shown in previous research by D’Alessandria et al. [55] and Bieda et al. [33]. For instance, temperature differences of 1400 K have been simulated in aluminum for spatial periods of 7.5 µm with a laser fluence of 1000 mJ∙cm^−2^ [55]. 

Finally, contact angle measurements were performed on the laser textured aluminum substrates. CAs were determined for samples fabricated with both spatial periods used (*Λ* of 8.0 µm and 19.0 µm), as well as fabricated with different laser processing parameters (pulse overlap and laser pulse repetition rate). However, since the CAs strongly correlated with structural parameters of the produced topographies, they have been correlated as a function of the structure height (h_St_), as shown in Figure 7 for the spatial periods *Λ* = 19.0 µm (Figure 7a) and *Λ* = 8.0 µm (Figure 7b). The error bars showed in the figures represent standard deviations of the result values averaged from five measuring points.

As can be seen, for the large interference period *Λ* = 19.0 µm (Figure 7a), considerable high water contact angles (even over 150°) were measured for samples that were treated using pulse repetition rates of 2 kHz and 5 kHz. For example, for the line-like patterns produced at a 2 kHz repetition rate, the contact angles increased slightly with increasing structure height, from 128° at h_St_ = 2.3 µm up to ~160° at h_St_ = 28.0 µm. For patterns produced with 5 kHz, the measured CAs were slightly higher compared to the structure produced at 2 kHz. For the patterns produced with a pulse repetition rate of 0.1 kHz, the contact angle decreased significantly with increasing structure height from 115° at h_St_ = 0.9 µm down to ~75° for h_St_ = 16.9 µm.

The differences in CA observed as a function of the rep. rate denote a change in the wettability state (Cassie–Baxter or Wenzel) which is not only influenced by the structure height but also by the pattern morphology, as will be demonstrated in the next section.

The contact angles investigated for the smaller interference period of 8.0 µm (Figure 7b) showed quite different correlations. The structures produced with a pulse repetition rate of 0.1 kHz showed an important increase in CA, even over 160°, for all measured conditions (e.g., independently of the structure height). For the structures manufactured at 2 kHz and 5 kHz pulse repetition rates, any clear trend was able to be observed, especially since for the same condition very different CAs were measured (hydrophobic or hydrophilic), which denoted a very unstable wettability state (see size of the error bars).

In order to understand the differences between both cases, CAs for both the Wenzel and Cassie-Baxter states were calculated for selected treated samples, as shown in Table 2. These samples were chosen based on their having similar measured structure heights but having been fabricated at different repetition rates. A comparison of the measured values with the theoretical calculated values shows that for all cases the CAs are similar to the values calculated for either the Wenzel or Cassie-Baxter wetting states. For example, for the spatial period *Λ* = 19.0 µm, for the treated aluminum sample at a 0.1 kHz repetition rate, the measured CA (73.8°) was very similar to the angle calculated for the Wenzel state (79.9°), whereas for the structures produced with 2 kHz and 5 kHz pulse repetition rates, the obtained CAs were similar to the Cassie-Baxter condition (2 kHz: measured = 151.7°, calculated = 144.4°; 5 kHz: measured = 154.9°, calculated = 145.4°). For the *Λ* = 8.0 µm periodic patterns, the surfaces irradiated at 2 kHz and 5 kHz repetition rates showed similar CAs to the Wenzel case (2 kHz: measured = 72.4°, calculated = 82.6°; 5 kHz: measured = 84.2°, calculated = 83.5°), whereas those produced at a 0.1 kHz pulse repetition rate correlated well with the Cassie-Baxter state (0.1 kHz: measured = 160.8°, calculated = 135.5°).

Consequently, the obtained results demonstrate that the structural information (structure height and period) obtained from the topography of the produced patterns can explain the reported CAs (see Equations (3) and (5)) but cannot clarify why certain surfaces adopt either Wenzel or Cassie-Baxter conditions. For this reason, we proceed to further investigate possible differences in the surface morphology of the treated areas, which cannot be observed using conventional confocal microscopy.

Figure 8 shows different high magnification SEM images of some of the aluminum samples which have been described in Table 2.

The structures produced with a spatial period of *Λ* = 8.0 µm at both a 0.1 kHz and 5.0 kHz pulse repetition rate (Figure 8a,b) present an additional surface roughness on top of the hills (corresponding to the minima position). This nanoroughness in combination with the DLIP topography (on the micrometer scale) forms a hierarchical surface pattern which in general is associated with a super hydrophobic condition facilitating the Cassie-Baxter state. This hypothesis is able to explain the super hydrophobic condition reached by the sample treated with 0.1 kHz (with a CA = 160.8°) but not that for the sample processed at 5 kHz (with a CA = 84.2°). The last sample (Figure 8b) is also characterized by having trenches between the hills which are narrower and not continuous and which are interrupted by so-called kissing-points, which are presumably caused by material collapse in the molten state due to the high pulse repetition rates used in the process. Due to the narrower hills (see inset in Figure 8b), the surface in contact with the water approaches an idealized flat surface, which can explain the lower contact angle (CA = 84.2°) for this condition.

For the *Λ* = 19.0 µm period patterns, the trenches are continuous and rather uniform for both the patterns produced at 0.1 kHz (Figure 8c) as well as 5 kHz pulse repetition rates (Figure 8d). In particular, the structures produced at a 5 kHz pulse repetition rate tend to form material overhangs on the ridges of the patterns, but, differently from the *Λ* = 8.0 µm textures, no kissing-point was observed and the separation distance between the overhangs is much larger (Figure 8d). Further, in the patterns produced at a 5 kHz pulse repetition rate (Figure 8d), the surface nano-roughness differs from that produced at a 0.1 kHz pulse repetition rate (Figure 8c), showing crystal-like nano-roughness (Figure 8d) or slurry-like surface texture (Figure 8c), respectively. In consequence, it is possible that the existence of the overhangs (without interconnections) in combination with the observed nano-roughness can explain the superhydrophobic properties for this condition (CA = 154.9° for 5 kHz), since, as is known from the literature, this can lead to increased contact angles because the liquid cannot not fill the gaps. Conversely, the sample treated at 0.1 kHz with a large spatial period *Λ* = 19.0 µm shows a very open topography which can be entirely wetted by the water droplet and thus reach the Wenzel state.

For the application of water-repellent surfaces, not only are hydrophobic surface properties required but also a high stability of the hydrophobic condition. For instance, superhydrophobic surfaces under normal conditions can become unstable under low temperatures or external pressure [56]. Under these conditions, the Cassie state transits to a metastable state or even to the Wenzel state, which deteriorates the superhydrophobicity [57,58]. Additional features at the nano- or microscale can contribute to the stabilization of a superhydrophobic surface due to the higher surface energy difference between the Cassie and the Wenzel states [57,58]. Changes in the water wetting characteristics of surfaces as a function of time has also been reported by other authors [59,60]. For instance, Kietzig et al. showed that after creating dual scale roughness structures by laser irradiation, different metal alloys initially exhibited superhydrophilic behavior but later became almost superhydrophobic or actually superhydrophobic [60]. This effect was explained by a combined effect of surface morphology (laser induced dual scale roughness structure) and surface chemistry [59]. 

## 4. Conclusions

In this study, the development of a strategy for maximizing the height of periodic surface patterns in aluminum substrates utilizing DLIP with ns pulses was introduced in order to control their surface wettability. The DLIP assembly, which splits a laser beam into two sub-beams and then interferes them on the substrate, forms a line-like laser spot, allowed the optimal utilization of the full available laser energy provided by high power ns-pulsed lasers. Our experiments have shown that heat accumulation of the consecutive energy input by laser pulses leads to a local heating of the substrate volume, which intensifies molten material flow and eventually increases the capability of the method to produce patterns with high periodic structures. For the patterns produced with large spatial periods (19.0 µm), remarkably high structure aspect ratios larger than one could be achieved for the first time in this material, while for the shorter periods (8.0 µm) an aspect ratio of about one could be obtained, which still exceeds the previously achieved values for DLIP with ns-pulsed lasers. The investigation of the applicability of the produced surface patterns to producing super hydrophobic surfaces showed that certain surfaces exhibit contact angles even over 160°. Furthermore, the measured contact angles were compared with theoretical calculations based on Wenzel and Cassie-Baxter models, showing a good correlation between the experiments and the calculations. It is important to mention that the decisive factor for producing super hydrophobic surfaces (corresponding to the Cassie-Baxter state) is not only determined by the height of the produced structures but also on their nanoroughness (on top of the patter hills) as well as the possibility to produce overhangs without kissing-points. These decisive topography characteristics depend strongly on the laser pulse repetition rate applied.

## Figures and Tables

**Figure 1 materials-12-01484-f001:**
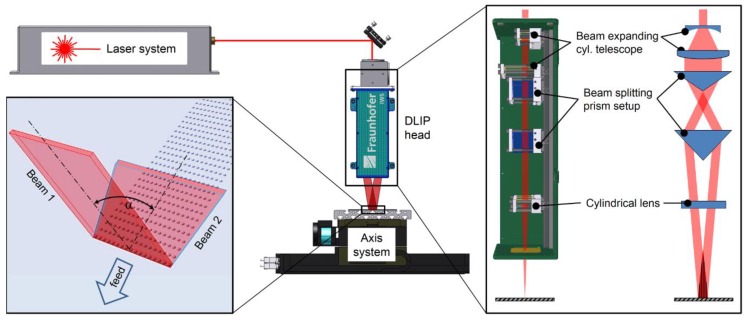
Modular optical assembly for high throughput direct laser interferential patterning (DLIP).

**Figure 2 materials-12-01484-f002:**
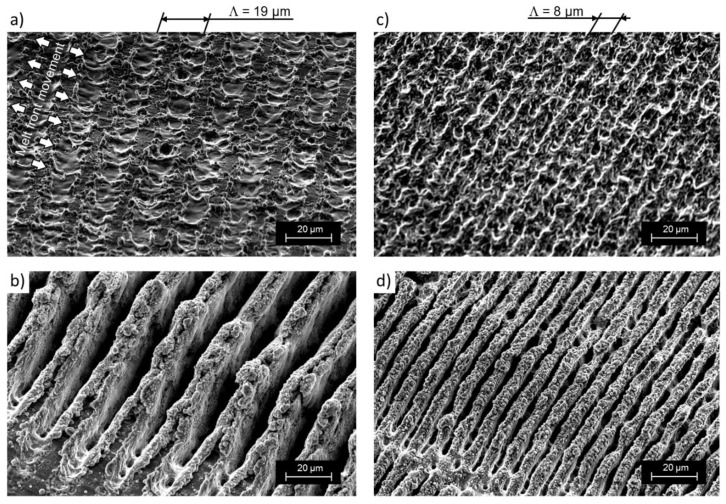
Scanning electron microscope images of DLIP structured aluminum surfaces with interference periods of (**a**,**b**) *Λ* = 19.0 µm and (**c**,**d**) *Λ* = 8.0 µm, fabricated using pulse overlaps (**a**,**c**) pulse overlap (PO) = 80% and (**b**,**d**) PO = 99%. The laser fluence was 3.2 J∙cm^−2^. Material: aluminum EN AW-5754 AlMg3 [43].

**Figure 3 materials-12-01484-f003:**
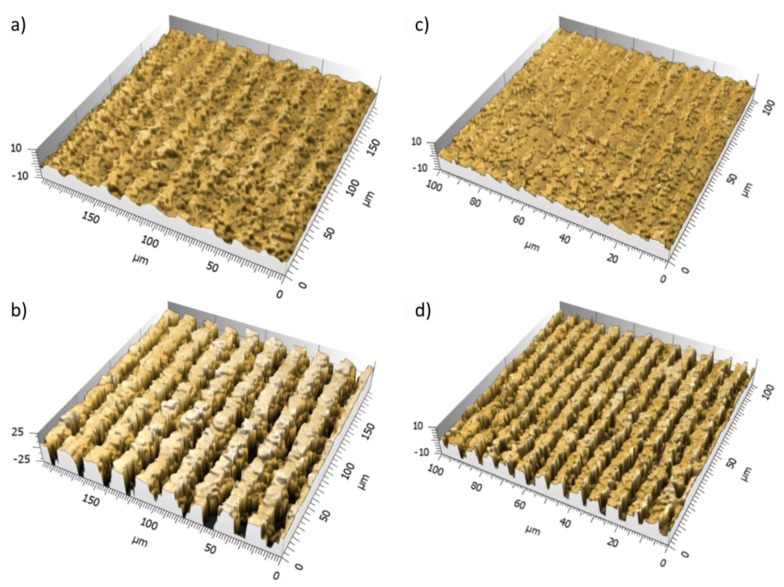
Confocal microscope images of DLIP structured aluminum surfaces with interference periods of (**a**,**b**) *Λ* = 19.0 µm and (**c**,**d**) *Λ*= 8.0 µm, fabricated using pulse overlaps (**a**,**c**) PO = 80% and (**b**,**d**) PO = 99%. The laser fluence was 3.2 J∙cm^−2^. Material: aluminum EN AW-5754 AlMg3 [43].

**Figure 4 materials-12-01484-f004:**
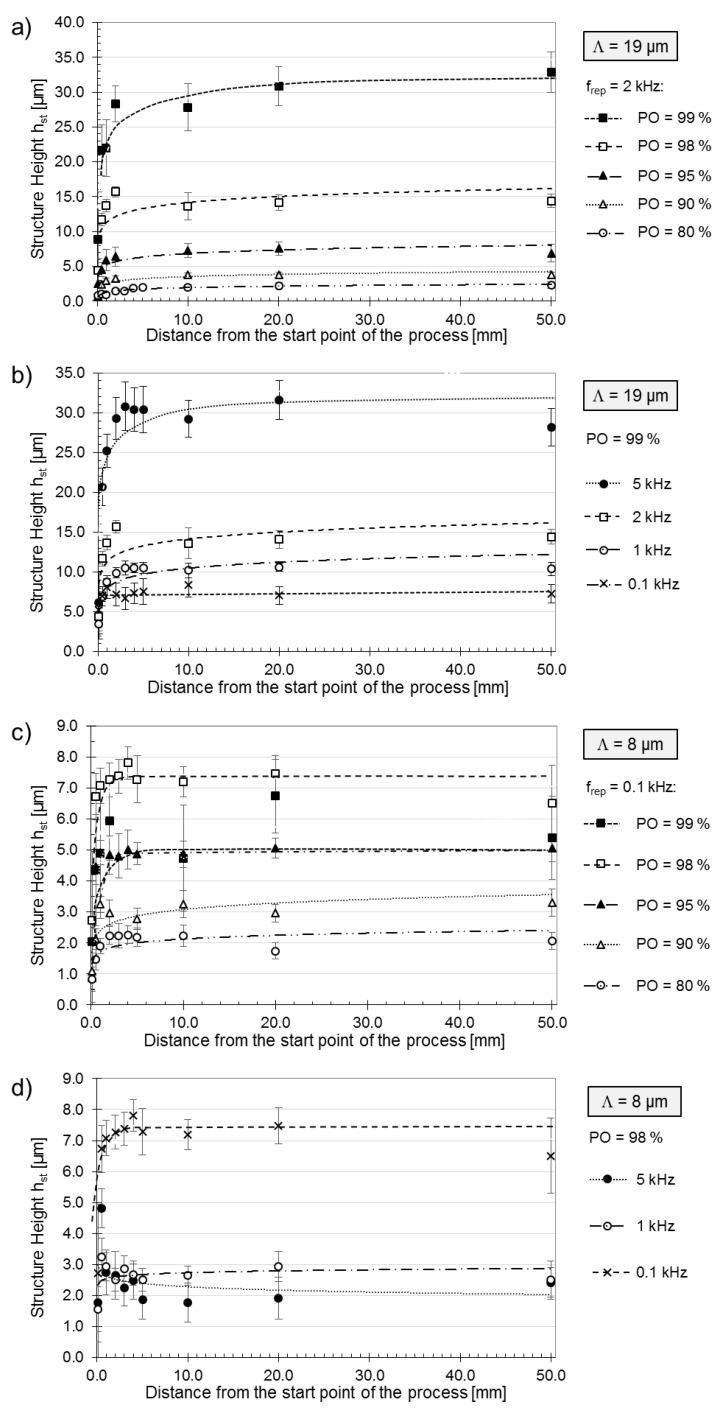
Evolution of the heights of trench patterns produced with DLIP: (**a**) *Λ*= 19.0 µm, variation of PO at fixed rep. rate f_rep_ = 2 kHz, (**b**) *Λ*= 19.0 µm, variation of rep. rate at fixed PO of 98%, (**c**) *Λ*= 8.0 µm, variation of PO at a fixed rep. rate f_rep_ = 0.1 kHz, and (**d**) *Λ*= 8.0 µm, variation of pulse repetition rate at fixed PO of 98%. Material: aluminum AlMg3. The laser fluence per pulse was 3.2 J∙cm^−2^.

**Figure 5 materials-12-01484-f005:**
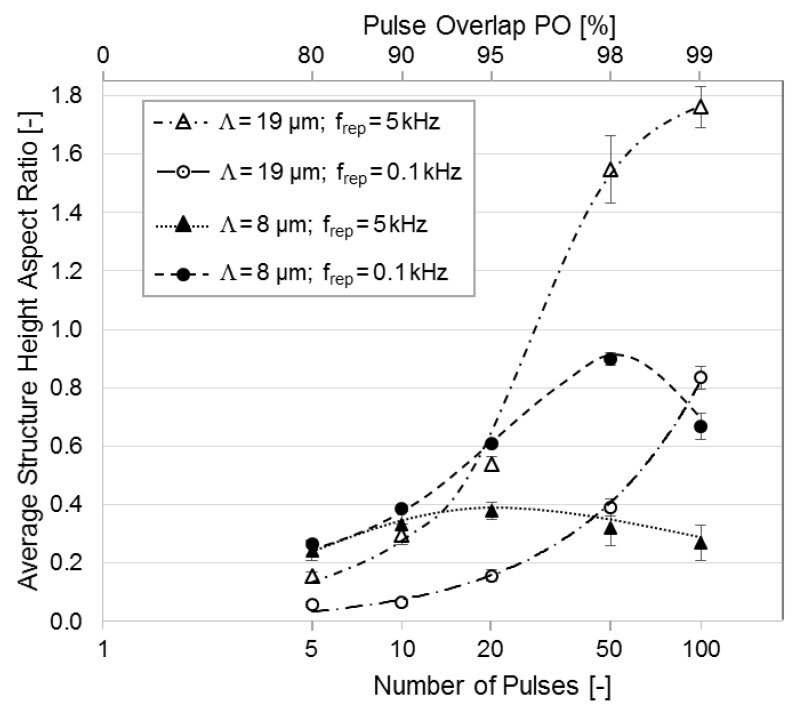
Average structure height aspect ratios for patterns with period *Λ* = 19.0 µm and *Λ* = 8.0 µm for different pulse repetition rates (f_rep_ of 0.1 and 5 kHz) depending on the pulse overlap. The applied laser fluence was 3.2 J∙cm^−2^. Dotted lines are only to guide the eyes; no uniform regression was applied.

**Figure 6 materials-12-01484-f006:**
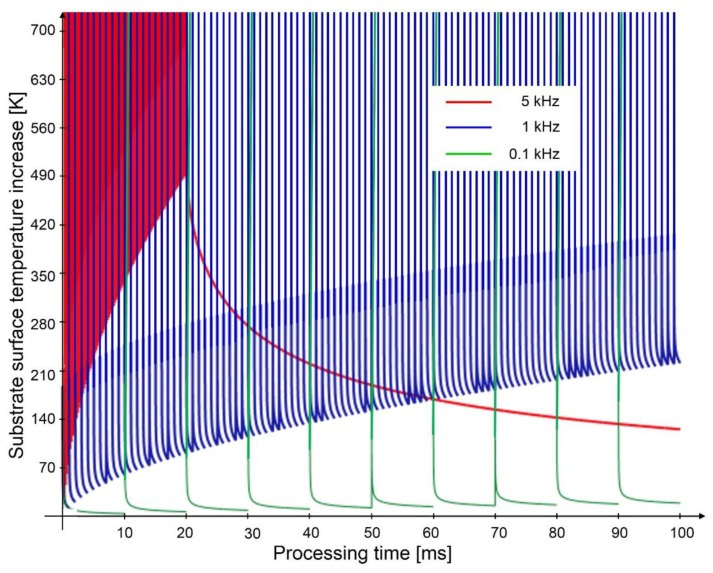
Calculation of the heating of a 1D volume of aluminum using a 10 ns laser pulse with a fluence of 3.2 J∙cm^−2^ and different pulse repetition rates of 0.1, 1, and 5 kHz.

**Figure 7 materials-12-01484-f007:**
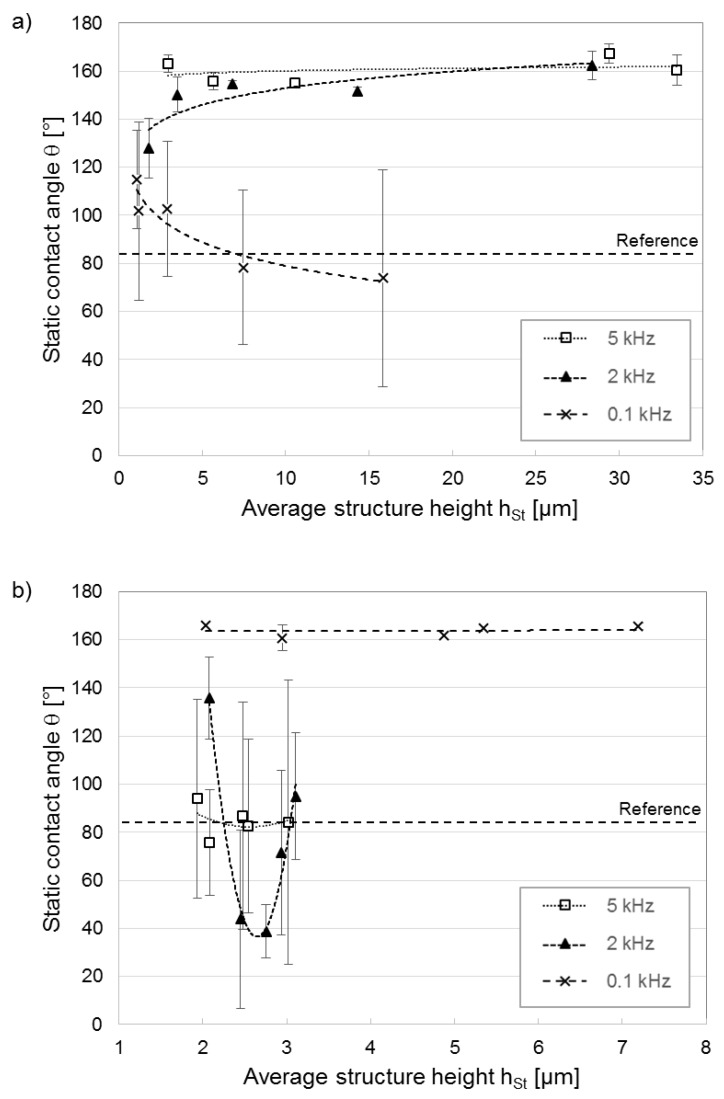
Static water contact angles as a function of structure depth and different process pulse repetition rates for (**a**) *Λ* = 19.0 µm and (**b**) *Λ* = 8.0 µm.

**Figure 8 materials-12-01484-f008:**
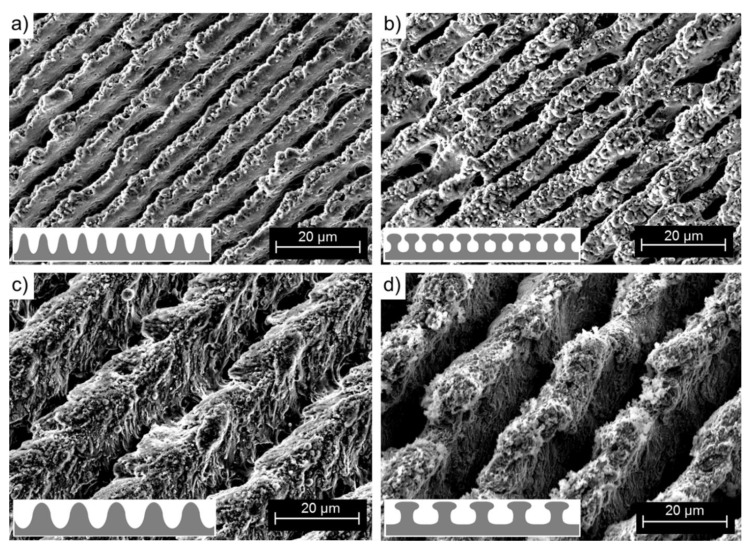
DLIP treated aluminum surfaces: (**a**) *Λ* = 8.0 µm, PO = 98%, repetition rate *f_rep_* = 0.1 kHz; (**b**) *Λ* = 8.0 µm, PO = 98%, repetition rate *f_rep_* = 5 kHz; (**c**) *Λ* = 19.0 µm, PO = 98%, repetition rate *f_rep_* = 0.1 kHz; and (**d**) period *Λ* = 19.0 µm, PO = 98%, repetition rate *f_rep_* = 0.1 kHz. Material: aluminum EN AW-5754 AlMg3 [43]. The used laser fluence per pulse was 3.2 J∙cm^−2^.

**Table 1 materials-12-01484-t001:** Material properties of aluminum EN AW-5754 AlMg3 [43].

Parameter	Symbol	Unit	Value
Density	*ρ*	kg∙m^−3^	2698.9
Thermal diffusivity	*κ*	m^2^·s^−1^	98.8 × 10^−6^
Specific heat capacity	*c_p_*	J∙(kg·K)^−1^	896

**Table 2 materials-12-01484-t002:** Comparison of selected measured contact angle (CA) with theoretical values calculated according to Wenzel and Cassie-Baxter.

Interference Period (µm)	Laser Pulse Rep. Rate (kHz)	Structure Height (µm)	Measured Contact Angle (°)	Calc. CA (Wenzel) (°)	Calc. CA (Cassie-Baxter) (°)
19.0	0.1	15.8	73.8	79.9	143.4
19.0	2.0	14.3	151.7	80.6	144.4
19.0	5.0	10.5	154.9	82.2	145.4
8.0	0.1	3.0	160.8	83.6	135.5
8.0	2.0	2.9	72.4	82.6	134.7
8.0	5.0	3.0	84.2	83.5	136.3

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
