# Peer review of "High Throughput Direct Laser Interference Patterning of Aluminum for Fabrication of Super Hydrophobic Surfaces"

_materials, 2019, doi:10.3390/ma12091484_

Reviewer 1 Report

Dear authors,

Congratulations for a valuable manuscript. In my opinion, you succeeded to present your work in a  proper organized way. You made your point regarding the theme concerning High Throughput Direct Laser Interferential Patterning of Aluminum for Fabrication of Super Hydrophobic Surfaces. I have no comments or suggestions for improving the content of your manuscript, I consider it is suitable for publishing. I recommended minor revision because of the following:          

Please check the following:

line 251, 252, 257, 269.Please check (Error! Reference source not found.

Same error: line 295, 298, 310, 311, 314, 315, 317.

Author Response

Response to Reviewer 1

Dear authors,

Congratulations for a valuable manuscript. In my opinion, you succeeded to present your work in aproper organized way. You made your point regarding the theme concerning High Throughput Direct Laser Interferential Patterning of Aluminum for Fabrication of Super Hydrophobic Surfaces. I have no comments or suggestions for improving the content of your manuscript, I consider it is suitable for publishing. I recommended minor revision because of the following:          

Please check the following:

line 251, 252, 257, 269.Please check (Error! Reference source not found.

Same error: line 295, 298, 310, 311, 314, 315, 317.

Answer:     We thank the reviewer very much for her/his comments. All errors were corrected.

Reviewer 2 Report

Major.

1.       Good  literature review of interference usage is given in the introduction of  the manuscript, however, 3D polymerization by interference lithography  is not included, for example, ref. [DOI: 10.2961/jlmn.2016.03.0013]. I  recommend include interference lithography into an overview of laser  interference applications.

2.       The  authors use two beam interference setup with interfering elongated  laser beam with THREE cylindrical lenses. Similar setup using only ONE  cylindrical lens and one spherical lens described in pioneering work  [DOI: 10.2961/jlmn.2010.01.0016]. The much simpler setup gives similar  results, therefore, I recommend clarify what is the benefit of usage of  the cylindrical telescope on the top of the experimental setup  (expansion of the beam in one direction?).

3.       The  authors use calculations with includes thermal properties of aluminum  (Table 1) for evaluation of thermal diffusivity, however, the number of  thermal diffusivity for Al (m^2/s) is missing in the paper, please  provide the thermal diffusivity number. The ratio between temperatures  in maxima and minima is calculated in work published by Lasagni et al. [DOI: 10.1109/3M-NANO.2012.6473000]. I recommend providing an estimation of the ratio of temperatures in maxima and minima in the current of your work (8-19 um period and 10 ns pulse duration).

4.       The  equation numbering is not in sequence (1), (4), (6), (7), (2), (3),  (4), (5), etc. Please correct and make equation numbers in sequence like  (1), (2), (3) ...

5.       The  melt front movement is shown in Figure 2, also Marangoni convection is  mentioned in the text. The thermal capillarity convection (also called  Marangoni effect) and its related Plateau-Rayleigh instability in  processing using a cylindrical lens can be theoretically evaluated by [http://dx.doi.org/10.1016/j.apsusc.2013.03.092].  The periodical spikes (cylindrical molted ridge formation to  periodically arranged droplets) are seen in Figure 2 (a) in the (Y)  direction perpendicular to inference periodicity (X) direction. However,  no theoretical evaluation of transverse periodicity (Y direction) by  physical numbers is given in the manuscript.  I recommend include some  discussion regarding the periodicity and evaluation of transverse  droplet period by measuring the radius of curvature of the ridge (period  should follow 9.02*R0, where R0 is the radius of curvature of the  molten ridge).

6.       The  measured and calculated by Cassie-Baxter and Wenzel models of contact  angles are given in table 2. I would be clearer to see in on the graph  how wettability models correspond to experimental data.

Minor.

7.       The parameter "ls" is not declared in equation (4).

              8.     The "Error! Reference source not found" are found in lines 295, 320, etc. Please insert correct hyperlinks.   

Author Response

Response to Reviewer 2

 1.      Good literature review of interference usage is given in the introduction of  the manuscript, however, 3D polymerization by interference lithography  is not included, for example, ref. [DOI: 10.2961/jlmn.2016.03.0013]. I  recommend include interference lithography into an overview of laser  interference applications.

Answer:      We are grateful to the reviewer for drawing our attention to the work described in the recommended reference. The research achievements described in that paper are undoubtedly significant. However, there are major differences in content to the work here submitted. In contrast to laser interference lithography, where photoresist is used as additional material and irradiated by the laser, in direct laser interference patterning the patterns are produced on the substrate surface by selective ablation and/or melting. However, in order to mention other possibilities to fabricate patterns using interfering beams, the following paragraph has been modified (see page 2 of new manuscript):

“…While in laser interference lithography a photoresist material is needed for producing the periodic structure, in direct laser interference patterning the patterns are produced on the substrate surface by selective ablation and/or melting [41].”

                                The following References have been added to the manuscript:

41.          Stankevičius, E.; Garliauskas, M.; Račiukaitis, G. Bessel-like Beam Array Generation Using Round-tip Micro-structures and Their Use in the Material Treatment. J. Laser Micro Nanoen. 2016, 11, pp. 352-356

 2.      The authors use two beam interference setup with interfering elongated laser beam with THREE cylindrical lenses. Similar setup using only ONE cylindrical lens and one spherical lens described in pioneering work  [DOI: 10.2961/jlmn.2010.01.0016]. The much simpler setup gives similar results, therefore, I recommend clarify what is the benefit of usage of  the cylindrical telescope on the top of the experimental setup  (expansion of the beam in one direction?).

Answer:      We thank the reviewer for advising us of the significant pioneering work described in the recommended reference. Please, note that we are also utilizing only one cylindrical lens and two prisms. The optical setup presented in our manuscript allows the dimensions of the focus of the interfering laser beams incident on the substrate to be specifically tailored to the requirements of the application, e.g. setting a suitable fluence or fulfilling industrial process specifications. The following text was added to the manuscript regarding to this additional reference in page 2:

“…A laser interference setup with elongated elliptical laser spots has been also developed by Molotokaite et al. to produce micro-surface patterns on thin metal films deposited on glass substrates [42].”   

                                The following References have been added to the manuscript:

42.          Molotokaite, E.; Gedvilas, M.; Račiukaitis, G.; Girdauskas, V. Picosecond Laser Beam Interference Ablation of Thin Metal Films on Glass Substrate. J. Laser Micro Nanoen. 2010, 5, pp. 74-79

 3.      The authors use calculations with includes thermal properties of aluminum  (Table 1) for evaluation of thermal diffusivity, however, the number of  thermal diffusivity for Al (m^2/s) is missing in the paper, please  provide the thermal diffusivity number. The ratio between temperatures  in maxima and minima is calculated in work published by Lasagni et al. [DOI: 10.1109/3M-NANO.2012.6473000]. I recommend providing an estimation of the ratio of temperatures in maxima and minima in the current of your work (8-19 um period and 10 ns pulse duration).

         Answer:      The thermal diffusivity is now given in Table 1.

                We thank the reviewer for this contribution to enrich the discussion in our work. The following text was added to the manuscript regarding to this additional reference in page 11:

                “…The differences between the temperatures at maxima and minima positions has a significant influence on pattern formation, as it has been shown in previous studies by D'Alessandria et el. [54] and Bieda et al. [55]. For instance, temperature differences of 1400 K has been simulated in Aluminum for spatial periods of 7.5 µm with a laser fluence of 1000 mJ∙cm-2 applied [54].”

The following References have been added to the manuscript:

54.          D'Alessandria, M.; Lasagni, A.F., Mücklich, F. Direct micro-patterning of aluminum substrates via laser interference metallurgy. Appl. Surf. Sci. 2008, 255, pp. 3210–3216

55.          Bieda, M.; Lasagni, A.F.; Beyer, E. Direct Fabrication of Hierarchical Microstructures on Metals by Means of Direct Laser Interference Patterning. J. Eng. Mater. Technol. 2010, 132, pp. 031015-1-031015-6

 4.      The equation numbering is not in sequence (1), (4), (6), (7), (2), (3),  (4), (5), etc. Please correct and make equation numbers in sequence like  (1), (2), (3)

Answer:     This error has been corrected.

5.      The melt front movement is shown in Figure 2, also Marangoni convection is  mentioned in the text. The thermal capillarity convection (also called  Marangoni effect) and its related Plateau-Rayleigh instability in  processing using a cylindrical lens can be theoretically evaluated by [http://dx.doi.org/10.1016/j.apsusc.2013.03.092].  The periodical spikes (cylindrical molted ridge formation to  periodically arranged droplets) are seen in Figure 2 (a) in the (Y)  direction perpendicular to inference periodicity (X) direction. However,  no theoretical evaluation of transverse periodicity (Y direction) by  physical numbers is given in the manuscript.  I recommend include some  discussion regarding the periodicity and evaluation of transverse  droplet period by measuring the radius of curvature of the ridge (period  should follow 9.02*R0, where R0 is the radius of curvature of the  molten ridge).

Answer:      We thank the reviewer for the literature recommendation, which reveals very interesting approaches about thermal capillarity convection.

However, the comparability to the work published by us is very limited for the following reasons. Approaches with self-organizing periodic elements, can achieve suitable surface modifications in certain areas, but differ fundamentally from laser interfering methods. In laser interference structuring, the period is specifically controlled by the angle of the overlapping laser beams. This possibility of active management is a key feature of laser interference technology. Furthermore, the laser source used by us did not provide a Gaussian beam, therefore the results of the proposed reference are not transferable to our work.

6.      The measured and calculated by Cassie-Baxter and Wenzel models of contact  angles are given in table 2. I would be clearer to see in on the graph  how wettability models correspond to experimental data.

Answer:      We agree with the reviewer that a representation of the calculated water contact angles in the diagram with the measured values is relevant to improve its comprehensibility. However, the diagram of measured values (Figure 7) already contains a multitude of data points. Since the focus in the diagram (Figure 7) is on the pattern height values, we consider that is better not to include all this information in the same diagram. The graphs will be less readable by adding further points. The selection of the calculated values was based on systematic criteria and is intentionally shown in a separate listing (Table 2). 

       7.      The parameter "ls" is not declared in equation (4).

Answer:      The description of the variables in the equation has been updated as         follows:

“…According to eq. (2), the pulse overlap is a function of the feed rate vf, the repetition rate frep and the spot size ls (in the direction of movement)...”

8.      The "Error! Reference source not found" are found in lines 295, 320, etc. Please insert correct hyperlinks.

Answer:     This error has been corrected.

Reviewer 3 Report

A very interesting paper about the fabrication of super hydrophobic surfaces by DLIP. The paper is well written and organized and the conclusions are supported by the data presented by the authors. From a scientific point of view, the paper could be published without changes but too many typos are present in the text. Many letters, especially (but not only) those indicating the interference spatial period, are missed in the text and many error codes, related to figures 5-8, are present too. In addition, reference [10] is incomplete and some journal titles in the references  are abbreviated while all the others are not.

Author Response

Response to Reviewer 3

A very interesting paper about the fabrication of super hydrophobic surfaces by DLIP. The paper is well written and organized and the conclusions are supported by the data presented by the authors. From a scientific point of view, the paper could be published without changes but too many typos are present in the text. Many letters, especially (but not only) those indicating the interference spatial period, are missed in the text and many error codes, related to figures 5-8, are present too. In addition, reference [10] is incomplete and some journal titles in the references are abbreviated while all the others are not.

Answer: We thank the reviewer for her/his kind approval of our work. All errors have been corrected. Reference [10] has also been revised. All journal titles are now abbreviated according to the instruction for authors of this journal.

Reviewer 4 Report

The paper "High Throughput Direct Laser Interferential Patterning of Aluminum for Fabrication of Super Hydrophobic Surfaces" is a thoroughly prepared manuscript, presenting interesting experimental findings which are interpreted in an adequate way. The paper is publishable after the minor revision.

The notion of the "water contact angle" used by the authors for characterization of the reported surfaces is misleading and confusing.  The notion of an "apparent contact angle should be used for the characterization" of the nano- and micro-rough and chemically heterogeneous surfaces, see:

Drelich J. W. Contact angles: From past mistakes to new developments through liquid-solid adhesion measurements, Advances  Colloid & Interface Science,  267 (2019) 1-14.

Bormashenko Ed. Wetting of real surfaces, de Gruyter, Berlin, 2018.

 Marmur A. A guide to the equilibrium contact angles maze, in Contact Angle Wettability and Adhesion, V. 6, pp.3-18, ed. by K. L. Mittal, VSP, Leiden, 2009.

2. Eq. 4 is not the Cassie equation. 

3. The discussion of the stability of the Cassie wetting state observed on the reported surfaces is desirable. 

4. The list of references should be expanded, much experience has been gained in thee field, see: Kietzig A. M., et al. Patterned Superhydrophobic Metallic Surfaces, Langmuir200925 (8), pp 4821–4827; Ketzig A. M. et al., Fabrication of Micro/Nano Structures on Metals by Femtosecond Laser Micromachining, Micromachines 20145(4), 1219-1253;

Author Response

Response to Reviewer 4

1.      The notion of the "water contact angle" used by the authors for characterization of the reported surfaces is misleading and confusing.  The notion of an "apparent contact angle" should be used for the characterization of the nano- and micro-rough and chemically heterogeneous surfaces, see:

Drelich J. W. Contact angles: From past mistakes to new developments through liquid-solid adhesion measurements, Advances  Colloid & Interface Science,  267 (2019) 1-14.

Bormashenko Ed. Wetting of real surfaces, de Gruyter, Berlin, 2018.

Marmur A. A guide to the equilibrium contact angles maze, in Contact Angle Wettability and Adhesion, V. 6, pp.3-18, ed. by K. L. Mittal, VSP, Leiden, 2009.

Answer:     We would like to thank the reviewer for this comment.

                    The term “water contact angle” (WCA) was replace by “apparent contact angle (CA)” along the manuscript[VL1] .

2.      Eq. 4 is not the Cassie equation.

Answer:     We would like to thank the reviewer for pointing out this issue. Indeed, the equation provided represents a special case of the Cassie-Baxter equation established by Marmur for surfaces with a line-like geometry. Accordingly, the text in the manuscript was changed on page 4 as follows:

“…For the case of liquid covering a fabric made of rounded fibers (which is similar to a line-like geometry), an special case of the Cassie-Baxter equation with the addition of a roughness factor (rl) was established by Marmur [45]:”.

3.      The discussion of the stability of the Cassie wetting state observed on the reported surfaces is desirable.

Answer:     After further research on this aspect, the text in the new version of the manuscript on page 13 was completed as follows:

“…For the application of water-repellent surfaces, not only hydrophobic surface properties are required, but also a high stability of the hydrophobic condition. For instance, superhydrophobic surfaces under normal conditions can become unstable under low temperature or external pressure [56]. Under these conditions, the Cassie state transits to a metastable state or even to the Wenzel state, which deteriorates the superhydrophobicity [57,58]. Additional features in the nano or microscale, can contributes to the stabilization of a superhydrophobic surface due to the higher surface energy difference between the Cassie and the Wenzel states [57,58].“

                      The following References have been added to the manuscript:

56.       Long, J.; Pan, L.; Fan, P.; Gong, D.; Jiang, D.; Zhang, H.; Li, L.; Zhong, M. Cassie-State Stability of Metallic Superhydrophobic Surfaces with Various Micro/Nanostructures Produced by a Femtosecond Laser. Langmuir, 2016, 32, pp. 1065−1072

57.       Su, Y.; Ji, B.; Zhang,K.; Gao, H.; Huang, Y.; Hwang, K. Nano to Micro Structural Hierarchy Is Crucial for Stable Superhydrophobic and Water-Repellent Surfaces. Langmuir, 2010, 26, pp. 4984–4989

58.       Whyman, G.; Bormashenko, E. How to Make the Cassie Wetting State Stable?. Langmuir, 2011, 27, pp. 8171–8176

4.      The list of references should be expanded, much experience has been gained in the field, see: Kietzig A. M., et al. Patterned Superhydrophobic Metallic Surfaces, Langmuir, 2009, 25 (8), pp 4821–4827; Ketzig A. M. et al., Fabrication of Micro/Nano Structures on Metals by Femtosecond Laser Micromachining, Micromachines 2014, 5(4), 1219-1253;

Answer:     We thank the reviewer for the suggestion of additional literature. After a new research, the manuscript was changed in page 14 as follows:

“…Changes in the water wetting characteristics of surface as a function of time has also been reported by other authors [59,60]. For instance, Kietzig et al. showed that after creating dual scale roughness structures by laser irradiation, different metal alloys initially exhibited superhydrophilic behavior, but later became almost superhydrophobic or superhydrophobic [59]. This effect was explained by a combined effect of surface morphology (laser induced dual scale roughness structure) and surface chemistry [59].”.    

The following References have been added to the manuscript:

59.       Kietzig, A.-M.; Hatzikiriakos, S. G.; Englezos, P. Patterned Superhydrophobic Metallic Surfaces. Langmuir, 2009, 25, pp. 4821-4827   

60.       Milles, S.; Voisiat, B.; Nitschke, M.; Lasagni, A.F. Influence of roughness achieved by periodic structures on the wettability of aluminum using direct laser writing and direct laser interference patterning technology. J. Mater. Process. Technol. 2019, 270, pp. 142-151

  [VL1]These References were not added to the manuscript, since they do not fit well

Round  2

Reviewer 2 Report

Accept for publication in Materials journal.